# Cone-Beam Computed Tomography: A New Tool on the Horizon for Forensic Dentistry

**DOI:** 10.3390/ijerph19095352

**Published:** 2022-04-28

**Authors:** Rakhi Issrani, Namdeo Prabhu, Mohammed Ghazi Sghaireen, Kiran Kumar Ganji, Ali Mosfer A. Alqahtani, Tamer Saleh ALJamaan, Amal Mohammed Alanazi, Sarah Hatab Alanazi, Mohammad Khursheed Alam, Manay Srinivas Munisekhar

**Affiliations:** 1Department of Preventive Dentistry, College of Dentistry, Jouf University, Sakaka 72345, Saudi Arabia; kiranperio@gmail.com (K.K.G.); sara.alanazi@jodent.org (S.H.A.); mkalam@ju.edu.sa (M.K.A.); dr.srinivas.manay@jodent.org (M.S.M.); 2Department of Oral & Maxillofacial Surgery and Diagnostic Sciences, College of Dentistry, Jouf University, Sakaka 72345, Saudi Arabia; drpranam@gmail.com; 3Department of Prosthetic Dental Sciences, College of Dentistry, Jouf University, Sakaka 72345, Saudi Arabia; dr.mohammed.sghaireen@jodent.org (M.G.S.); dr.amal.alanazi@jodent.org (A.M.A.); 4Department of Periodontics, Sharad Pawar Dental College & Hospital, Datta Meghe Institute of Medical Sciences, Nagpur 440001, India; 5Department of Diagnostic Dental Sciences, College of Dentistry, King Khalid University, Abha 62529, Saudi Arabia; al.alqahtani@kku.edu.sa; 6Dental Department, Armed Forces Hospital, King Abdel Aziz Airbase, Dhahran 34641, Saudi Arabia; jamaan73@hotmail.com; 7Center of Transdisciplinary Research (CFTR), Saveetha Dental College, Saveetha Institute of Medical and Technical Sciences, Saveetha University, Chennai 600077, India; 8Department of Public Health, Faculty of Allied Health Sciences, Daffodil International University, Dhaka 1207, Bangladesh

**Keywords:** dentistry, forensic, teeth, investigations, deceased

## Abstract

Teeth and bones of calvarium are important structures from a forensic point of view, as they are extremely resilient to destruction or decomposition, even under temperature variations. Radiology is inevitably an important tool in forensic investigations. Maxillofacial radiology provides a considerable amount of information for the identification of remains and evidence in case of legal matters. The advent of cone-beam computed tomography (CBCT) in the arena of maxillofacial 3D imaging has contributed immensely to forensic science such as the age estimation through teeth, analysis of bite marks, determination of race and sex, etc. The advantages of accuracy in imaging the anatomy, digitized technology favoring easier comparison of records and storage of records for a longer period, cost reduction, dose reduction, and easier portability have made it an unavoidable adjunct in forensic investigations. The aim of this paper is to review and highlight the importance of CBCT in successful forensic identification and analysis. This review is written to address the various aspects of CBCT as a recently developed technology that may be very useful in some forensic contexts, based on searches for current studies in the literature using PubMed, Scopus, Web of Science, and Google Scholar databases, to identify studies published since inception to December 2021, with no language restriction. In conclusion, CBCT is an accessible 3D imaging technology with many applications, one of them being in forensic sciences.

## 1. Introduction

Bishop Oscar Romero quoted *“those who have a voice must speak for those who are voiceless”* [1]. Hence, forensic science, which uses physical, functional, psychic, normal, or pathological characteristics to establish the identity of the unidentified and deceased, plays an important role for those who are voiceless. The process of forensic identification uses a systematic comparison of acquired postmortem (PM) data with antemortem (AM) data of the missing/deceased person. Bones of the maxillofacial region and teeth are one of the strongest structures of the human body, as they can endure the taphonomic processes and destruction for ages even under chemical and temperature variations [2].

Radiology is an inevitable tool for forensic science. It aids in the analysis of anatomical structures for personal identification, estimation of biologic age, assessment of abuse or assault, and identification of weapon of assault but also comes in handy for comparison of AM and PM records in case of forensic investigation. In 1896, Prof. Arthur Schuster used radiology for the first time in forensics to demonstrate the lead bullets in the head of a victim [3]. To overcome the limitations of 2D radiography such as the dimensional variations in image, dissimilar vertical and horizontal angulation, etc., 3D imaging in forensic identification is necessary [4]. The development of computed tomography (CT) by Sir Godfrey Hounsfield in 1967 [5] and magnetic resonance imaging (MRI) by Paul Lauterbur in 1973 [6] ushered in the era of 3D imaging. Postmortem facsimile imaging can be constructed by using the information provided in AM CT scans [7]. Cone-beam CT (CBCT) scanners, first built by Robles RA in 1982 [8] for angiography, have now become an important mode of 3D imaging, especially in dentistry. In this article, we aim to review the contribution of CBCT to the forensic identification process.

## 2. Cone-Beam Computed Tomography

CBCT is a revolutionary discovery that is being widely used in all fields of dentistry, including orthodontics, endodontics, oral surgery/pathology, periodontics, and implant treatment planning [9]. It utilizes a cone-shaped X-ray beam and a single, flat-panel detector or image intensifier radiation detector to produce multiple multiplanar images in a sequential manner as the X-ray source and detector fixed on a platform rotate 180–360 degrees around the object, which are then mathematically reconstructed into a volumetric dataset that is converted into a readable image by Feldkamp algorithm [9]. For 3D volumetric analysis, the data obtained by CBCT can also be transferred into third-party image enhancing software such as Mimics (Materialise NV, Leuven, Belgium) and 3-Matics (Materialise NV, Belgium), which have been used in various forensic applications [10]. CBCT has the advantages of less severe metallic artifacts, low image reconstruction times, easier portability, and dose reduction (96% lower than conventional CT) [11]. Thus, CBCT has become the gold standard for imaging the oral and maxillofacial region.

### Radiographs in Forensic Dentistry

Radiographs play important roles in forensic examinations, as they provide objective evidence of the anatomical conditions and the dental treatments provided up to that point in time. They are beneficial due to their ease, simplicity, and quick modes of obtaining information in a non-destructive manner. Furthermore, they are economical in comparison to DNA technology [12,13].

Dental identification is achieved by comparing the AM and PM radiographs of dental and maxillofacial structures and determining the concordant points for positive or possible identification [12]. These data are assessed based upon many dental findings—the teeth present or missing, supernumerary teeth, crown and root structure of a tooth, pulpal anatomy, decayed teeth, restored teeth, endodontic treatment prosthetic treatment, to name a few [11,12]. The presence of sufficient characteristic features between the AM and PM data that are identical with explainable differences is considered a positive identification [14].

CBCT allows a precise examination of the spatial relationships of dental structures such as teeth, roots, and supporting structures on AM and PM images, in addition to facilitating correct alignment between AM and PM radiographs without the need for new exposures. Additionally, it assists in reconstructive technology to form a biological profile for a missing/deceased person with an unknown identity [15,16]. Table 1 shows the comparison of imaging systems in forensic odontology [17,18,19,20,21].

The areas to which CBCT has contributed regarding forensic identification are described in subsequent sections.

## 3. Age Estimation

Age determination of human remains aims to assist in establishing age in case of litigation or when creating a biological profile of the deceased individual. Volumetric analysis of dental features in radiographs such as reduction in pulpal size, which leads to the formation of secondary dentin, aids in age estimation since secondary dentin deposition is proportional to the age of an individual [22]. This correlation was first explained by Bodecker in 1925 [23]. It is categorized into the following three phases [22]:

*(A) Prenatal, neonatal, and postnatal*: This phase is based on the prenatal jawbones, the appearance of tooth germs, early mineralization of primary teeth during intrauterine life, and the extent of crown completion;

*(B) Children and adolescents*: This phase is based on the appearance of tooth germs, earliest detectable trace of mineralization of teeth, degree of crown completion, eruption time of a tooth in the oral cavity, the extent of root completion of erupted and/or unerupted teeth, the extent of root resorption of deciduous teeth, and presence of open apices in teeth;

*(C) Adult*: This includes pulp volume assessment of teeth and formation of the third molar.

Based on the morphological changes in teeth, several methods have been proposed to determine the age in adults—namely, the following:

(a) Assessment of tooth-pulp volumes

Age estimation by assessment of secondary dentin deposition was first introduced by Gustafson [24] in 1950. In 1995, Kvaal et al. [13] were the pioneers in the development of an indirect age estimation method based on the amount of secondary dentin deposition using radiographic images. By surveying mandibular premolars and molars with visible pulp cavities on panoramic radiographs, they also suggested different methods for transverse and longitudinal measurement of the teeth using simple linear regression analysis to establish the relationship between the pulp–tooth index and chronological age [13]. The most preferred teeth for age estimation are maxillary canines. They are preferred owing to their characteristic features such as being single-rooted teeth with the largest pulp area, usually retained till later in life, less susceptible to caries, with less wear, compared with posterior teeth, and hence, easiest to analyze [25]. Multirooted teeth are usually not studied for age estimation because the pulp changes are less evident in the root, although they are clear in the canal. Additionally, they are more susceptible to caries and being missed or damaged due to wear [25].

Two-dimensional (2D) imaging is limited to measurements only in the vertical and mesiodistal dimensions of pulp chambers. Factors such as malpositioning, rotations, crowding, magnification or foreshortening, and superimpositions make the analysis of 2D radiographs difficult and inaccurate. CBCT enables the use of a third dimension, i.e., buccolingual measurements for calculating the pulp tooth ratio, by providing dimensionally accurate images with easy tools for measurements and calculation [26].

Enamel is generally resistant to alterations beyond normal wear and tear; conversely, the pulp–dentin complex displays physiologic and pathological changes with progressing age. Characteristically, to quantify these changes, extraction and sectioning of teeth are necessary, which is not always a practicable choice. CBCT, however, affords a non-invasive substitute [26].

Yang et al. [26] were the pioneers in the correlation of the pulp–tooth volume ratio with age from CBCT scans of 28 different single-rooted teeth. They used a custom-made voxel counting software for this purpose and found a moderate correlation with a linear relationship between pulp–tooth volume ratio and age [26]. Jagannathan et al. (2011) [27] studied 188 extracted mandibular canines using volumetric reconstruction of their CBCT images to evaluate the suitability of pulp–tooth volume ratio for age estimation in an Indian subpopulation. They concluded that the pulp–tooth volume ratio is a useful indicator of age, although correlations may vary in different populations, necessitating the application of population-specific formulae for the same [27]. Biuki et al. (2017) [28] used CBCT scans to examine the pulp–tooth volume ratio in single-rooted teeth and made use of a primitive custom-made software program. They designed a formula to estimate the age and found an inverse and significant correlation between the pulp–tooth volume ratios and age in males and females, with a stronger correlation in males than females, indicating that gender affects the formula used to estimate age. The correlation was stronger in maxillary central incisors and canines. They also found that the simultaneous use of maxillary canines gave a stronger and better age assessment, compared with the examination of all of the four canines [28]. Furthermore, Kazmi et al. (2019) [29] found a statistically significant difference in volumes of pulp between males and females and that mandibular canine pulp volume and sex had the highest predictive power in estimating the age of deceased person.

(b) Third molar analysis

The development of the third molar, which usually occurs after 17 years of age, is used as a guide to assess the age of an individual [22]. Third molars are the most varied teeth in terms of their anatomical features, agenesis, and age of eruption [30]. Harris and Nortje (1984) [31] described the five stages of root development of the third molar, correlating with the mean ages and mean length. Van Heerden (1985) [32] also described the five stages by evaluating the development of the mesial root of the third molar using panoramic radiographs. Asif et al. (2019) [10] observed a strong inverse correlation between the chronological age and surface area of the developing mandibular third molar apices in CBCT images.

For assessing the age in the context of forensics, third molar development is used in the age range of 14–21 years, when all other teeth have completed their formation. As per the recommendations of the Study Group on Forensic Age Diagnostics, the Demirjian method has the greatest predictive and practical value. This method of age estimation is advantageous since the stages of development are clearly defined, according to radiographs, diagrams, and written criteria [30].

(c) Assessment of spheno-occipital synchondrosis

Spheno-occipital synchondrosis (SOS) is the site of union of the sphenoid and occipital bone that is situated in the clivus area at the base of the skull, anterior to the foramen magnum and inferior to the pituitary fossa. It is a skeletal–anthropological marker and is the principal growth cartilage of the cranial base during childhood that is active up to the age of 12–15 years. SOS is the last site in the cranium to terminate growth, which usually occurs by 20 years of age [33].

Sinanoglu et al. (2016) [34] analyzed SOS using CBCT scans among 7–25-year-old subjects and noted that the mean ages for the complete fusion of SOS were 18 and 20 for females and males, respectively. They proposed that CBCT images could be a method of choice for age assessment for the aforementioned age group and can also aid in determining the age of 18 years, which is important for legal issues involving the establishment of adulthood of a person. The legal age reflects a presumption that typical individuals of that age are “mature enough to function in society as adults, to care for themselves, and to make their own self-interested decisions”. Every country and/or state has adopted a legal age of majority through various legislative or judicial measures [35].

Sobh et al. (2020) [36] visualized SOS using CBCT in a mid-sagittal view in a neutral head position to create statistically quantified age estimation standards in a sample of the Egyptian subpopulation. Their study concluded that all subjects with non-fused SOS were minors (<18 years). Sharma et al. (2020) [37] analyzed SOS in coronal and sagittal planes with CBCT in the Indian subpopulation, and its closure was graded in six stages. This study depicted a linear relationship between the closure of SOS and age while observing that males tend to attain each stage later than females, and this sexual dimorphism exists till the age of 16 years; thereafter, 100% population shows complete closure of SOS.

Suture closure might be influenced by many factors such as nutritional and health status, exercise, and physical activities of an individual, general growth, allometric growth and development of the bones, and, to some extent, race. Due to the wide variability in the age of closure, as proposed by various researchers, it provides a general pattern at various age levels [37]. Therefore, SOS is considered to be a more reliable tool for age estimation when used in conjunction with other age assessment tools such as third molar mineralization, cervical vertebrae maturation, etc. [37].

## 4. Sex Determination

Sex determination is essential to approximate the biological profile of an individual. Bones that are conventionally used for sex determination are the pelvis, skull, and long bones. The accuracy rate of sex determination is found to be 100% from a skeleton, 98% from both the skull and pelvis, 95% from pelvis only or the pelvis and long bones, 90–95% from both the skull and the long bones, and 80–90% from the long bones only [20]. Anthropometric measurements of various craniofacial structures such as the mandible, mastoid process, foramen magnum, sinuses, and nasal septum on CBCT images can be used for assessing the sex of an individual [2].

(a) Mandibular measurements

Mandible exhibits sexual dimorphism in terms of its shape and size, which can be used to estimate sex with a high degree of accuracy [38,39]. The bone thickness and size of the female skeleton are smaller than those of the male attributed to sex, nutrition, and physical activity. Due to the difference in the masticatory forces between the males and females, the relative development (such as size, strength, and angulation) of the masticatory muscles is found to affect the expression of mandibular dimorphism [40]. Males have a forward rotation in the mandible, whereas females have been found to have downward and backward rotation in the mandible, making the gonial angle values higher in females than in males. The gonial angle changes during life—at birth, it is rather obtuse, then decreasing as one grows up and increasing again in old age [39]. CBCT can be useful for various measurements of the mandible for gender determination such as ramus length and breath, gonion–gnathion length, gonial angle, bigonial breadth, and bicondylar breadth. Okkesim et al. (2020) [41] conducted a retrospective study in which they reconstructed CBCT images of the mandible for coronoid height, condylar height, ramus height, and maximum and minimum ramus breadth as parameters for determining gender and found that all variables of the mandibular ramus on CBCT models showed a statistically significant difference among the sexes. Gopal et al. (2016) [40] retrospectively studied 100 CBCT to analyze the same parameters and demonstrated that the ramus can be used in the determination of sex even when the complete mandible was not retrievable for forensic studies.

(b) Foramen Magnum

Foramen Magnum (FM) is a 3D aperture that lies within the basal central region of the occipital bone and is a transition zone between the spine and skull [42]. The cranial base is supposed to be a useful guide for forensic examination because it is comparatively thick, compact, and protected due to its anatomical position, so this part of the skull tends to withstand physical insults [2]. Various studies have indicated that a statistically significant difference exists between males and females for foramen length, breadth, and area. Teixeria (1982) [43] first reported that the measurements related to the FM can be a useful tool in the sex estimation of an individual. Gunay and Altinkok (2000) [44] assessed the area of FM in an inhomogeneous sample of male and female skeletons and reported that the mean value of FM area in males is higher than that in females, but this is insufficient for forensic determination of sex. Tambawala et al. (2016) [45] established that CBCT had an accuracy rate of 66.4% for sex prediction in forensic assessment of FM on reformatted axial sections.

(c) Paranasal sinuses

Paranasal sinuses (PNS) remain intact, even upon high physical injury to the skull and other craniofacial bones [46]. Various studies have shown that the jaw sinuses area unit is considerably larger in males than in females [2]. Culbert and Law (1927) [47] conducted the first radiographic comparison of PNS in the body. Wanzeler et al. (2019) [48] performed a volumetric analysis of the maxillary, sphenoidal, and frontal sinuses using CBCT scans and co-related them with FM measurements. They concluded that by summing up the volumes of maxillary, sphenoidal, and frontal sinuses, the chances to correctly determine an individual’s sex are 96.2% and 92.7% for males and females, respectively, and on correlating the sum of the three estimated PNS with FM measurements, sex identification chances increased to 100% [48].

i. Frontal sinus comparison

Frontal sinus (FS) assessment is a primary tool for personal identification, but the significant sexual dimorphic characteristics in its measurements can also be used for gender determination [2]. They are unique in every individual, even monozygotic twins. They show no change after the age of 20 years until old age when atrophic changes might lead to gradual pneumatization [49]. The peculiar morphological features of the frontal sinus made positive identification possible from burnt fragmented remains of a US journalist who disappeared in Guatemala in March 1985 [50].

The frontal sinus index (FSI) (height/width ratio) is a tool that is usually used for sex determination [2]. The points analyzed on lateral cephalometric radiographs for this purpose are the nasion–sella line horizontally [2]. Other parameters that have been used in CBCT analysis of frontal sinuses are FS width and height, anteroposterior diameter, total sinus width, inter-sinus width, the distance between highest points between two the FS, and distance between the highest point of left FS and maximum lateral limit measurements [51]. Benghiac et al. (2015) [52] assessed FSI on CBCT scans and concluded that the predictive value of FSI for determining gender may need to be supplemented with other information to achieve a high level of accuracy. Choi et al. (2018) [53] used reconstructed CBCT scans to measure different parameters of FS and reported the sexual dimorphism with an accuracy of 80%.

ii. Maxillary sinuses

The first paranasal sinus to develop is the maxillary sinus (MS). It appears at the end of the second embryonic month, and its development is completed by the age of 18 to 20 years [45]. A CBCT study by Tambawala et al. (2016) [54] showed that the males had statistically significantly higher values for both the left and right maxillary sinuses in terms of height, width, and length dimensions, with maxillary sinus height being the best predictor of sexual dimorphism. Similar findings were reported by Paknahad et al. (2018) [55], who found a 76% overall accuracy rate of sex determination. Urooge et al. (2017) [46] evaluated CBCT scans to examine the maxillary sinus for different parameters of MS and showed that the efficiency of MS to identify gender was 74% in females and 68% in males, with an overall accuracy of 71%.

(d) Mastoid process

The anatomical position of the mastoid region at the base of the skull makes it resistant to injury, and therefore, it remains intact even if the skull is fractured; this makes the mastoid region an area of interest for macroscopic identification of sex [56]. There are anatomical differences in the mastoid process among males and females; for example, the tip of the mastoid process is vertical in males and faces inward in females, and males have larger mastoid measurements than females [37,56]. Amin et al. (2015) [57] conducted a study on 192 CBCT skull images to examine the size, surface area, flare, and medial convergence angle of the mastoid process and observed that these features of the mastoid process can correctly identify the sex in 90.6% of the subjects. Farhadian et al. (2020) [56] assessed 190 CBCT skull images for the following varied landmarks related to the mastoid process and noted a statistically significant difference between sexes for most of the variables.

(e) Others

CBCT has been a useful tool in gender determination and personal identification using various other anatomical landmarks such as orbital apertures [2], volumetric assessment of the dental crowns [58], the morphology of the articular eminence and the mandibular fossa [59], mandibular midline neurovascular canal structures [60], and shapes of cervical vertebrae [61], to name a few.

## 5. Implant Backtracking

The contemporary era of dental management has made dental implants unique identifiers with extreme importance as forensic evidence for human identification. The specific designs of dental implants such as perforations, grooves, top chambers, and threads, which become visible solely at certain rotations or angulations, make them a unique forensic identification tool. A panoramic radiographic study by Gopal et al. (2017) [62] for implant backtracking showed an accuracy of 82.5%, paving the way for CBCT studies on the same for personal identification [2].

## 6. Bite Mark Analysis

With the advent of CBCT, bite marks can now be turned into more consistent evidence. Wu et al. (2013) [63] compared the overlays of suspected dental casts of bite marks with CBCT images and plate scanning, which automatically generated comparison overlays. The blind identification of two kinds of bite marks data illustrated that CBCT used in bite mark identification had very high specificity, thus making it an accurate and effective tool, compared with the conventional method. Marques et al. (2013) [64] used CBCT to study bite marks in foodstuffs that might be found in a forensic case scenario and found it possible to perform a metric analysis of bite marks in appropriately dense foodstuffs.

## 7. Determination of Facial Soft Tissue Thickness (FSTT) and Reconstructive Identification

Reconstructive identification is considered when no putative identification or AM records are available, especially in cases where the remains have been macerated or burnt beyond recognition. It encompasses the coordinated approach of forensic dentistry, forensic medicine, anthropology, anatomy, and 3D imaging (CT, CBCT, and MRI) [16]. The reconstruction of the lost soft tissue using the soft tissue pegs placed at specific sites on the face that represent the average FSTT is performed manually and by using computer-aided techniques for reconstructive identification [65]. The computerized method uses a laser video camera interfaced with a computer or with CT scanning. This method allows imaging of the skull data as a fully shaded 3D surface wherein the face can be drawn with the help of computer software [14,66].

Farman and Scarfe (2006) [67] stated that the assessment of CBCT images provides sufficient clarity in soft tissue definition that aids in determining the air/soft boundaries and the patient’s profile. Fourie et al. (2010) [68] aimed to determine the reliability and accuracy of CBCT scans in measuring the soft tissue thicknesses of the face by comparing the soft tissue thickness in cadaver heads at 11 craniofacial landmarks, physically with a dermal biopsy punch with CBCT using 0.3 and 0.4 mm resolution. They obtained a very high inter- and intra-observer correlation between both methods, hence establishing the reliability of CBCT for measuring FSTT with 0.3 mm resolution.

## 8. Evaluation and Demonstration of Cranial Trauma and Projectile Injuries

Radiologic methods have been used to examine the fracture patterns in blunt-force trauma of the human calvaria [12]. In the case in which a putative weapon is found, the weapon can be compared with the impressed injuries in the human skull [12]. The cases of firearm projectile injuries are usually difficult to examine due to the varied pathways of the bullet that might either not be in the body or be diverted by an anatomical structure. Therefore, knowing the location of the firearm projectile prior to the autopsy might facilitate the forensic examination [69]. Thus far, CT has remained the preferred modality for imaging projectiles. Every firearm projectile combination is associated with a typical injury pattern, thus making it possible to narrow down the likely weapon. However, high-density projectiles cause severe artifacts in CT, thus making it difficult to examine the anatomic structures in close proximity to the projectile [70]. Von See et al. (2009) [71] used CBCT and CT to analyze tissue destruction and the location of three different modern projectiles fired into the heads of pig cadavers. They concluded that CBCT is far superior to CT in detecting structural hard-tissue damage in the immediate vicinity of high-density projectiles, as it is less severely affected by metallic artifacts. Similar conclusions were drawn by Stuehmer C et al. (2019) [70] who studied viscerocranial gunshot injuries in 14 patients using CT and CBCT.

## 9. Discussion

CBCT has two main areas of use in forensic dentistry—diagnostic examination for the typing in the database of missing/deceased persons and the identification of individuals, especially casualties in criminal investigations and mass disasters [17]. The modern era has seen CBCT as a promising tool that provides a low-cost solution for undistorted 3D images of good quality with a low radiation dose while ensuring complete osseous anatomical description and exact localization [41]. Multiplanar correlation of the obtained images enables the operator to view the entire depth of the tissue with the additional advantage of 3D reconstruction with the help of sophisticated yet easy-to-use software. The small size and portable nature of CBCT make it suitable for field use, which is a boon in recent times, an era in which it is becoming increasingly common to conduct forensic investigations outside the traditional laboratory and morgue settings, especially in disasters and mass fatality investigations. Their design for self-calibration eliminates the need to reset the machine in case of movement into a location [72].

While CBCT scans were initially used exclusively for imaging the craniomaxillofacial complex, the plethora of research in this field has expanded its role in varied applications in the head and neck region, e.g., visualization of sinus pathologies, tumors of the skull base, 3D analysis of the upper airway dimensions of patients with obstructive sleep apnea or cleft lip and palate, visualization of cochlear implants, identifying fractures and intra-operative positioning of fixation plates [73]. Its superiority to CT in imaging injuries caused by high-density projectiles due to lesser artifacts has definitely made it a remarkable tool in forensic imaging. Recognizing its role in PM imaging, it has also experimentally been used at the Detroit Museum of Art, in the study of an Egyptian mummy and in the investigation of artifacts [72]. This ever-expanding scope of applications of clinical CBCT indicates that AM CBCT scans will be increasingly accessible, and it will eventually become an essential part of routine maxillofacial and otolaryngology investigations that may be useful for forensic identification if needed. As a matter of fact, presently, US officers leaving for international missions in areas at risk receive a CBCT prior to departure to serve as an AM identification dataset [60].

Few disadvantages of CBCT include limited soft-tissue contrast, compared with CT because of scattering, motion artifacts, and their inability to perform bone density measurements using Hounsfield units [2]. Furthermore, dual-energy scanning, which enhances tissue depiction, has yet to be studied using CBCT [73]. As CBCT is used primarily for examining craniofacial structures, currently, it only accommodates cranial and extremity imaging, and therefore, whole-body imaging using CBCT is not possible [73]. Differences across populations and ethnicity also necessitate more population-based studies with CBCT for age and gender determination, to provide reference material for different ethnic and racial groups and the ones currently used as baseline reference material needs to be updated. More studies should be carried out on implant backtracking using CBCT to develop a dataset that can serve as an AM record. With the increasing emphasis on virtual autopsies, there is a need to investigate the role of CBCT in them and develop dedicated tools and software programs to assist with procedures in virtual autopsies. Nevertheless, the scope and potential of this novel technology have still not been fully explored in the field of forensic investigations.

## 10. Conclusions

CBCT is undoubtedly emerging as a remarkable 3D imaging modality with varied applications. It offers several technical and practical advantages, in addition to being cost-effective, thus emphasizing that forensic investigators should consider using it more frequently for postmortem imaging. There is considerable scope for further research investigating the role of CBCT in various dental and forensic applications, making it an invaluable asset with many feathers in its cap.

## Figures and Tables

**Table 1 ijerph-19-05352-t001:** Comparison of imaging systems in forensic odontology.

Radiological Method	Forensic Odontology and Anthropology	Advantages	Disadvantages	Radiation Dose	Spatial Resolution
Cone-beam computed tomography	Generated panoramic image	Precise, single tooth evaluation detailed panoramic image skull volumes	Metal alloy artifacts	19–368 µSv 20 s scan 68 µS	Spatial resolution in a “best possible” experimental scenario of <3 lp mm^−1^ with a median value of approximately 2 lp mm^−1^
Multislice computed tomography–medical CT	Skeletal findings examination	Digital autopsy	3D virtual models	lower jaw 1320 µSv upper jaw 1400 µSv bimaxillary 2100 µSv	0.5–0.625 mm in the *z*-axis, and approximately 0.5 mm in the *x*- to *y*-axes
Magnetic Resonance Imaging	Valuable supplement to postmortem CT for the detection of wound channel and soft tissue injuries	Evaluation of soft tissue injuries	Metal alloy artifacts	--	1–2 mm for most sequences

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
