# Peer review of "Cone-Beam Computed Tomography: A New Tool on the Horizon for Forensic Dentistry"

_ijerph, 2022, doi:10.3390/ijerph19095352_

Round 1

Reviewer 1 Report

This article is devoted to an overview of non-destructive methods of investigation for the needs in the field of forensic science. Namely, the purpose of this paper is to highlight the importance of applying the Cone Beam Computed Tomography (CBCT) method in successful forensic identification and analysis. The paper is well organized and the overview includes modern articles from the last 5 years. It was made a comparison with other methods of data visualization and the authors had concluded that CBCT  has become the gold standard for imaging in the field of the oral and maxillofacial investigation. In addition, the authors emphasize the possibility of reconstructing the biological profile of the investigated  object, which is also a promising area.

However, in this paper also is describing the shortcomings of the new technology, which consist in the limiting of the soft tissue contrast, compared with the CT method, which makes it impossible to calculate bone density. Currently, imaging of the whole body using CBCT method is limited and requires further research and technical improvement. The advantage of this method, the authors point out, is small size and portable nature, which provides an inexpensive solution for 3D visualizing with good quality and a low radiation dose, while providing a complete bone anatomical description and accurate localization.

It will be interesting to mention the resolution of the applied research methods (CT, MRI, CBCT) for their more substantive comparison. And also, it should be noted that CBCT method is widely used not only in the field of forensic science, but also for the diagnosis of teeth, any part of the maxillofacial area in the orthopaedic and orthodontic fields.

Author Response

Dear Respected Reviewer,

Thank you for your valuable feedback/ suggestions.

Warm Regards

Reviewer 2 Report

Some considerations about the text.

In general, I consider that it is an interesting revision of the state of the art. Nevertheless I think there are some points that could be improved:

Line 97:

You must consider that the third molar is not always present.

Line 141: 

Could it be a good indicator to determine if an individual is older or younger than 18 years old?

this is interesting given that in many countries that 18 is the age of majority. I think you should discuss this, as it is an exciting fact within the forensic field.

Lines 145-146:

This would be related to the degree of dental wear of the third molars. Don't you think that dental wear can be greatly affected by the type of diet, for example? I think this is highly debatable and should be discussed in the article.

Line 176:

I would say that sex determination is essential to approximate the individual's biological profile, but "to identify" I think it is too ambitious term in this case. The identification is given by many other more specific and individualizing data of the individual.

Author Response

(The authors gave the same response as above.)

Reviewer 3 Report

Overall, a decent manuscript. I congratulate the authors for the same.

  1. Abstract- is well written but needs to be elaborated more.
  2. Line 50- heading should be bold same as section heading.
  3. Methods- mention of review methodology lacking-There is no mention of how the data was collected, from which databases it was collected, the inclusion and exclusion criteria, from which years, keywords used for search etc.
  4. Methods- Line 44,45,47- pls remove comma from the year.
  5. It would be nice to have a couple of schematic illustrations or images.
  6. Pls include acknowledgement, COI etc.

Author Response

(The authors gave the same response as above.)
